# The effect of air-pollution and weather exposure on mortality and hospital admission and implications for further research: A systematic scoping review

**Mary Abed Al Ahad**[1]\*, **Frank Sullivan**[2], **Urška Demšar**[1], **Maya Melhem**[3], **Hill Kulu**[1]

**1** School of Geography and Sustainable Development, University of St Andrews, Scotland, United Kingdom, **2** School of Medicine, University of St Andrews, Scotland, United Kingdom, **3** Department of Landscape Design and Ecosystem Management, American University of Beirut, Beirut, Lebanon

\* maaa1@st-andrews.ac.uk

**Data Availability Statement:** All relevant data are within the paper and its Supporting Information files.

## Abstract

### Background

Air-pollution and weather exposure beyond certain thresholds have serious effects on public health. Yet, there is lack of information on wider aspects including the role of some effect modifiers and the interaction between air-pollution and weather. This article aims at a comprehensive review and narrative summary of literature on the association of air-pollution and weather with mortality and hospital admissions; and to highlight literature gaps that require further research.

### Methods

We conducted a scoping literature review. The search on two databases (PubMed and Web-of-Science) from 2012 to 2020 using three conceptual categories of "environmental factors", "health outcomes", and "Geographical region" revealed a total of 951 records. The narrative synthesis included all original studies with time-series, cohort, or case cross-over design; with ambient air-pollution and/or weather exposure; and mortality and/or hospital admission outcomes.

### Results

The final review included 112 articles from which 70 involved mortality, 30 hospital admission, and 12 studies included both outcomes. Air-pollution was shown to act consistently as risk factor for all-causes, cardiovascular, respiratory, cerebrovascular and cancer mortality and hospital admissions. Hot and cold temperature was a risk factor for wide range of cardiovascular, respiratory, and psychiatric illness; yet, in few studies, the increase in temperature reduced the risk of hospital admissions for pulmonary embolism, angina pectoris, chest, and ischemic heart diseases. The role of effect modification in the included studies was investigated in terms of gender, age, and season but not in terms of ethnicity.

**Funding:** This review is part of a PhD project that is funded by the St Leonard's PhD scholarship, University of St Andrews, Scotland, United Kingdom. The open access publication fees were funded by the University of St Andrews Libraries, Scotland, United Kingdom.

**Competing interests:** The authors declare that they have no conflict of interest.

## Conclusion

Air-pollution and weather exposure beyond certain thresholds affect human health negatively. Effect modification of important socio-demographics such as ethnicity and the interaction between air-pollution and weather is often missed in the literature. Our findings highlight the need of further research in the area of health behaviour and mortality in relation to air-pollution and weather, to guide effective environmental health precautionary measures planning.

## Introduction

Air-pollution and weather exposure beyond region-specific thresholds have serious effects on the public health [1, 2]. Worldwide, population growth, increased urbanization, economic and industrial growth, intense energy consumption, high usage of transportation vehicles, improved living standards, and changing lifestyles and consumption patterns for at least the last 100 years have resulted in increased emissions of air pollutants including greenhouse gases; and fluctuations in ambient temperature and other weather variables [3, 4].

Ambient air-pollution consists of a range of pollutants including particulate matters with diameters of less than 10 μm (PM10) and less than 2.5 μm (PM2.5), nitrogen oxides (NOx) including nitrogen dioxide (NO2), Sulphur dioxide (SO2), Carbon monoxide (CO), and Ozone (O3) that have been associated with a range of different acute and chronic health conditions [5, 6].

Weather exposure in terms of changing temperature, relative humidity, rainfall and other weather patterns can cause a wide range of acute illness and result in deaths especially among vulnerable populations who lack adequate physiological and behavioural responses to weather fluctuations [7, 8]. Age (elderly and children vs adults), sex, socioeconomic factors (poverty, education, and ethnicity among others), pre-existing chronic diseases, use of certain medications, and environmental conditions such as the absence of central heating increase individual's susceptibility to environmental exposures [1, 9, 10]. Research has shown that hospital admissions and mortality increase when weather exposure exceed certain thresholds with lags up to 20 days [11–14].

Most of the literature has shown positive correlations of air-pollution and/or exposure to weather variables beyond region-specific thresholds with all-cause and cause-specific mortality and/or hospital admission especially related to respiratory and cardiovascular diseases [14–21]. Though, there is a lack of information on wider aspects including the role of some effect modifiers such as ethnicity and the interaction between air-pollution and weather factors. Literature has shown that ethnic minorities often live in more disadvantaged, highly populated urban communities with poor housing conditions and higher levels of air pollution exposure [22–24]. This results in poorer health and higher risk for chronic health problems with time. Similar to ethnicity, the interaction between air-pollution and weather variables in relation to health outcomes is often missed in the literature despite its importance in minimizing biased estimations. Air pollutants are highly reactive, and their formation is either catalysed or slowed down based on the existing weather conditions. For example, the presence of sunlight catalyses the formation of ozone pollutant resulting in higher ozone concentrations during the summer [25].

In this context, a thorough literature review is needed to map the available literature and highlight areas that require further research and investigation. Not to mention that further understanding of the effect of air-pollution and weather exposure on mortality and hospital admission is needed to achieve better environmental and health system planning, organization, resources allocation, and interventions. This article aims to provide a comprehensive review and narrative summary (not numerical estimate) of literature on the association of air-pollution and weather with mortality and hospital admissions; and to shed the light on areas that require further research. As far as we are aware, this is the first literature review examining the effect of multiple exposures (air-pollution and weather) on multiple outcomes (mortality and hospital admissions). We chose to focus our scoping literature review on countries that are part of the single European Union (EU) market (Austria, Belgium, Bulgaria, Croatia, Republic of Cyprus, Czech Republic, Denmark, Estonia, Finland, France, Germany, Greece, Hungary, Ireland, Italy, Latvia, Lithuania, Luxembourg, Malta, Netherlands, Poland, Portugal, Romania, Slovakia, Slovenia, Spain and Sweden, Norway, and Switzerland) and United Kingdom (UK) because these countries exhibit similar socio-economic, environmental, and health policies; minimizing the contextual differences in the effect of air-pollution and weather on mortality and hospital admission. Literature examining the effect of air-pollution and/or weather on mortality and hospital admissions in countries outside the EU and UK will be used for comparison purposes.

## Materials and methods

### Search strategy and database sources

To ensure methodological reliability, we carried out our scoping literature review according to the "Preferred Reporting Items for Systematic Reviews and Meta-Analyses for scoping reviews" (PRISMA-ScR) guidelines (S1 Checklist) [26].

A literature search was performed on the 6th of February 2020 using "PubMed" and "Web of Science" database sources that cover health, medical, and environmental literature. We attempted to assess the effects of air-pollution and weather events on mortality and hospital admission in Europe by searching original research articles published in peer-reviewed journals in the last 8 years (between 06/02/2012 and 06/02/2020 inclusive). We chose to review research published in the last 8 years because in March 2007, the European Union (EU) Heads of State and Government endorsed an "integrated climate change and energy strategy" that will come into action post the expiry of Kyoto Protocol targets in 2012 and that aims to combat climate change and weather fluctuations and cut air-pollution emissions to 30% below the 1990 levels [27].

Our search strategy was divided into three conceptual categories: "environmental factors", "health outcomes", and "Geographical region". The "Environmental factors" refers to air-pollution, including PM10, PM2.5, NO2, SO2, CO, and O3 air pollutants and to weather variables, including air temperature, rainfall, wind, relative humidity, and vapour pressure. The "health outcomes" include hospital admissions and mortality and the "Geographical region" refers to the EU countries and UK. For each conceptual category, a set of "MeSH" and "All Fields" terms joined by the Boolean operator "OR" were developed. Later, the three conceptual categories' search terms were joined using the Boolean operator "AND". Our search strategy excluded the "influenza infections", as these are considered confounders rather than outcomes for air-pollution and weather exposure. For more details about the search codes used to navigate PubMed and Web of Science search engines, please refer to S1 Table.

To minimize finding irrelevant literature, our search was limited to the following categories in the "Web of Science" search engine: environmental sciences, public environmental

occupational health, medicine general internal, environmental studies, multidisciplinary sciences, geosciences multidisciplinary, respiratory system, geography physical, geography, cardiac cardiovascular systems, urban studies, healthcare sciences services, peripheral vascular disease, medicine research experimental, emergency medicine, critical care medicine, health policy services, primary healthcare, social sciences biomedical, and demography. Grey literature, non-English language articles, conference abstracts, books, reports, masters and PhD dissertations, and unpublished studies were excluded from this review.

## Inclusion and exclusion criteria

To determine the studies that would be included in this scoping review, a set of inclusion and exclusion criteria were developed for the procedure of title, keyword, and abstract screening.

The inclusion criteria involved original quantitative research studies conducted in the EU and UK; that included at least one analysis where mortality and/or hospital admission was the outcome and where one or more of the following exposures were investigated: 1) ambient air pollutants including PM10, PM2.5, CO, NO2/NOx, SO2, and O3; 2) weather exposures including temperature, rainfall, wind, humidity, and pressure; and 3) extreme weather events including heat waves, cold spells, and droughts. Due to the large amount of literature on this topic and to allow comparable results between the studies, this review was limited to cohort, time-series, and case-crossover/self-controlled quantitative study designs where hazard ratios (HR), relative risks (RR), odd ratios (OR), or percentage increase were reported for quantifying the factors associated with mortality and hospital admission. These three study designs allow a temporal follow up to evaluate the effect of time varying exposures (air-pollution and weather) on the mortality and hospital admission health outcomes.

The exclusion criteria included the following:

- Methodological studies

- Original data studies that investigated the effect of ambient air-pollution and/or weather on mortality and/or hospital admission in countries outside the EU market and UK

- Articles studying the effect of indoor air-pollution on mortality and hospital admission

- Studies examining air-pollution and weather exposure on animals and plants

- Studies on occupational air-pollution exposure

- Non-English language articles

- Mortality and/or hospital admission projections and forecasting studies

- Protocol and letter to editor papers

- Qualitative research studies

- All types of literature reviews including but not limited to narrative, scoping, and systematic literature reviews

## Screening and data abstraction

Our search strategy revealed 487 articles from the "PubMed" database and 517 articles from the "Web of Science" database. These articles were exported to the citation manager software "Endnote" where 53 duplicates were identified and removed resulting in a total of 951 articles (Fig 1). Using the titles, key words, and abstracts, the 951 articles were screened for relevance according to the inclusion and exclusion criteria, explained in the previous section only by

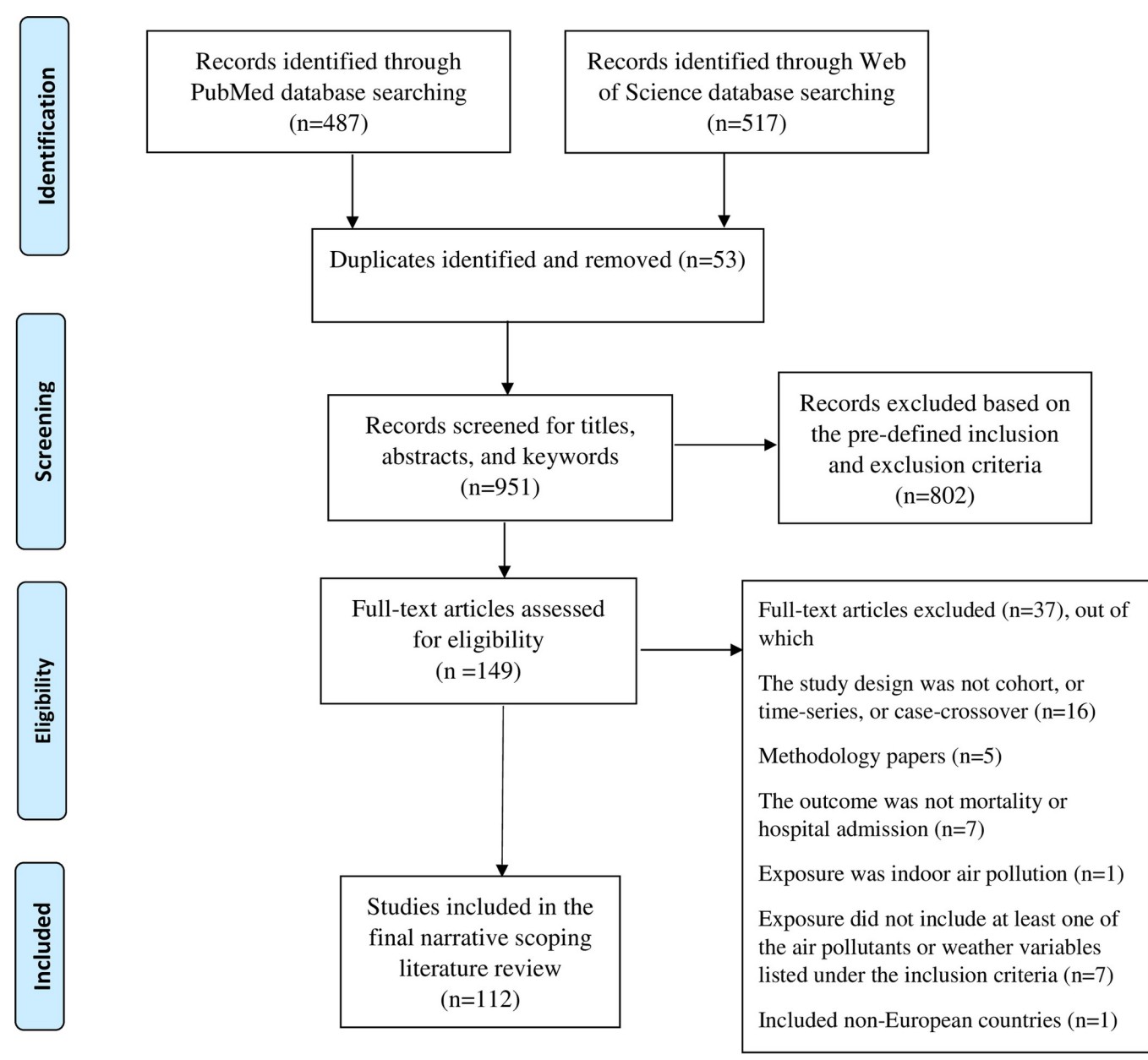

**Fig 1. PRISMA flow diagram illustrating the literature search, screening process, results, and exclusions.**

first author (MA). To ensure a rigorous and reliable application of the inclusion and exclusion criteria in the screening process, a second researcher (MM) screened independently a sample of 20% of the titles and abstracts of the 951 identified records. Disagreements between the two researchers were resolved through discussion until consensus was reached. All the studies that met the inclusion criteria (n = 149 articles) were retrieved for full text screening by MA. Following the full text screening phase, an additional 37 articles were excluded by MA resulting in a total of 112 articles to be included in the final narrative synthesis (Fig 1).

For narrative synthesis, the following information was retrieved from the 112 articles:

• Study design

- Location of the study population

- The outcome of interest

- Sample size

- Exposure variables

- The confounders adjusted for

- The assessed exposure time and the lags considered

- The exposure assessment method

- The statistical/modelling approach

- The relative risks (RR)/incident relative risks (IRR)/odd ratios (OR)/hazard ratios (HR) with their respective confidence intervals or the percentage increase that quantify the association between the outcome of interest (mortality and/or hospital admission) and the exposures (air-pollution and/or weather events).

## Ethical approval

Not applicable for this scoping literature review as it only includes descriptive narrative analysis of 112 published articles.

## Results

A total of 112 studies (S2 Table) were included in the final narrative review from which 70 involved the mortality outcome, 30 the hospital admission outcome, and 12 studies included both health outcomes (Table 1). Most of the studies used the time-series study design (n = 74, 66%) with Poisson models for data analysis, while minority of the reviewed studies employed the case-crossover design (n = 19, 17%) with conditional logistic regression for data analysis, and the cohort design (n = 18, 16%) with Cox hazard regression for data analysis (Table 1).

Most of the studies examined all-cause, cardiovascular and respiratory disease mortality and hospital admission outcomes while some studies tried to focus more directly on certain types of specific diseases such as psychiatric disorders including mania and depression, pulmonary embolism, myocardial infarction, stroke, ischemic heart disease, arrhythmias, atrial fibrillation, heart failure, cerebrovascular disease, chronic obstructive pulmonary disease (COPD), lung cancer, and diabetes (Table 1).

Table 1 shows the descriptive statistics of the included articles. S2 Table summarise the characteristics of the included studies in more details by the type of investigated health outcome. S3 Table demonstrate the included article's reported associations in terms of coefficients with 95% confidence intervals between air-pollution and/or weather exposure and mortality and/or hospital admission outcomes.

## The effect of air-pollution on mortality and hospital admission

In this review, six air pollutants (PM2.5, PM10, O3, CO, SO2, and NO2/NOx) were identified as causes of increased rates of mortality and hospital admissions. Each pollutant affects a range of diseases, most commonly, cardiovascular, respiratory, and cerebrovascular diseases. Some of the health effects can be immediate while others might appear after several days of initial exposure (Fig 2).

**Table 1. Descriptive characteristics of the included articles (N = 112).**

| Characteristics | Number of studies | Percentage |
|---|---|---|
| *Study design* | | |
| Cohort | 18 | 16% |
| Time series | 74 | 66% |
| Case-crossover | 19 | 17% |
| self-controlled case-series | 1 | 1% |
| *Study follow up time* | | |
| <5 years | 16 | 14% |
| 5 to 10 years | 48 | 43% |
| >10 years | 48 | 43% |
| [a]*Exposures* | | |
| PM10 | 50 | 45% |
| PM2.5 | 29 | 26% |
| O3 | 19 | 17% |
| NO2/NOx | 40 | 36% |
| SO2 | 12 | 11% |
| CO | 10 | 9% |
| Other air pollutants | 7 | 6% |
| Temperature | 56 | 50% |
| Humidity | 4 | 4% |
| Rainfall | 7 | 6% |
| Other weather variables | 10 | 9% |
| *Outcome in general* | | |
| Mortality | 70 | 63% |
| Hospital admission | 30 | 27% |
| Both: mortality and hospital admission | 12 | 11% |
| [a]*Specific Outcomes* | | |
| All-causes | 69 | 62% |
| Cardiovascular | 51 | 46% |
| Respiratory | 44 | 39% |
| Cerebrovascular | 10 | 9% |
| Cancer | 4 | 4% |
| Psychiatric disorders | 6 | 5% |
| Pulmonary embolism | 2 | 2% |
| Myocardial infarction | 9 | 8% |
| Stroke | 6 | 5% |
| Ischemic heart disease | 11 | 10% |
| Arrhythmias | 2 | 2% |
| Atrial fibrillation | 1 | 1% |
| Heart failure | 5 | 4% |
| Angina pectoris | 2 | 2% |
| Chronic obstructive pulmonary disease (COPD) | 6 | 5% |
| Asthma | 1 | 1% |
| Diabetes | 1 | 1% |
| Sudden infant death | 1 | 1% |
| Other types of illness | 13 | 12% |

[a]Percentages do not add up to 100% as categories are not mutually exclusive.

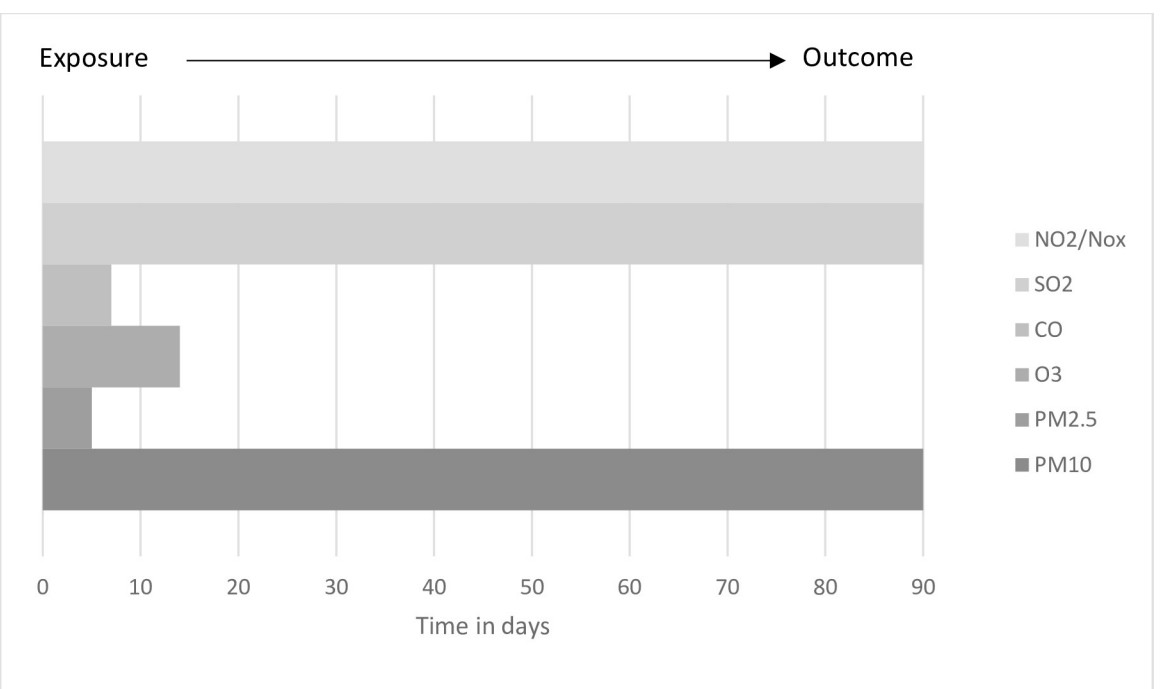

**Fig 2. The time range for health outcomes following exposure by type of air pollutant.**

**The effect of particulate matter pollutants on mortality and hospital admission.** Particulate matter is a heterogeneous mixtures of liquid droplets and solid particles suspended in the air that can result either from natural resources (windblown Saharan and non-Saharan dust, volcano ashes, forest fires, pollen, etc. . .) or from man-made activities including industrial processes, transportation vehicle smoke, burning of fossil fuels, extensive energy usage, combustion processes, and grinding and mining industries [28]. Due to its size, mass composition, and chemical components, particulate matter with larger diameter (PM10) will be deposited in nasal cavities and upper airways while particulate matter with smaller diameter (PM2.5) may penetrate more deeply the respiratory system reaching the alveoli and blood stream, carrying with them various toxic substances [29]. This in turn will cause health problems in humans such as asthma, irregular heartbeat, nonfatal heart attacks, decreased lung function, coughing and difficulty breathing symptoms [30].

Our review showed that PM10 air-pollution is positively associated with a range of cardiovascular and respiratory diseases mortality and hospital admission outcomes (Fig 3A and S3 Table). Fischer et al. (2015) showed an elevated hazard of 1.06 (95% CI = 1.04 to 1.08) for cardiovascular disease mortality for every 10 μg/m3 increase in PM10 pollution in the Netherlands [31]. Likewise, PM10 pollution acted as a risk factor for respiratory diseases mortality (HR = 1.11, 95%CI = 1.08 to 1.15; RR = 1.056, 95%CI = 1.043 to 1.069) [21, 32] and hospital admission (%increase = 0.69, 95% CI = 0.20 to 1.19) [20].

Air-pollution with PM2.5 exhibited a similar effect on human health as that of PM10 (Fig 3B and S3 Table). Nevertheless, PM2.5 was shown to have a greater risk on human health as compared to PM10 due to its smaller diameter size allowing more deep penetration into the respiratory system [33]. In France, Sanyal et al. (2018) showed an increased risk of 1.11 and 1.02 for all-cause hospital admission and moratality respectively per 10 μg/m3 increase in PM2.5 pollutant [32].

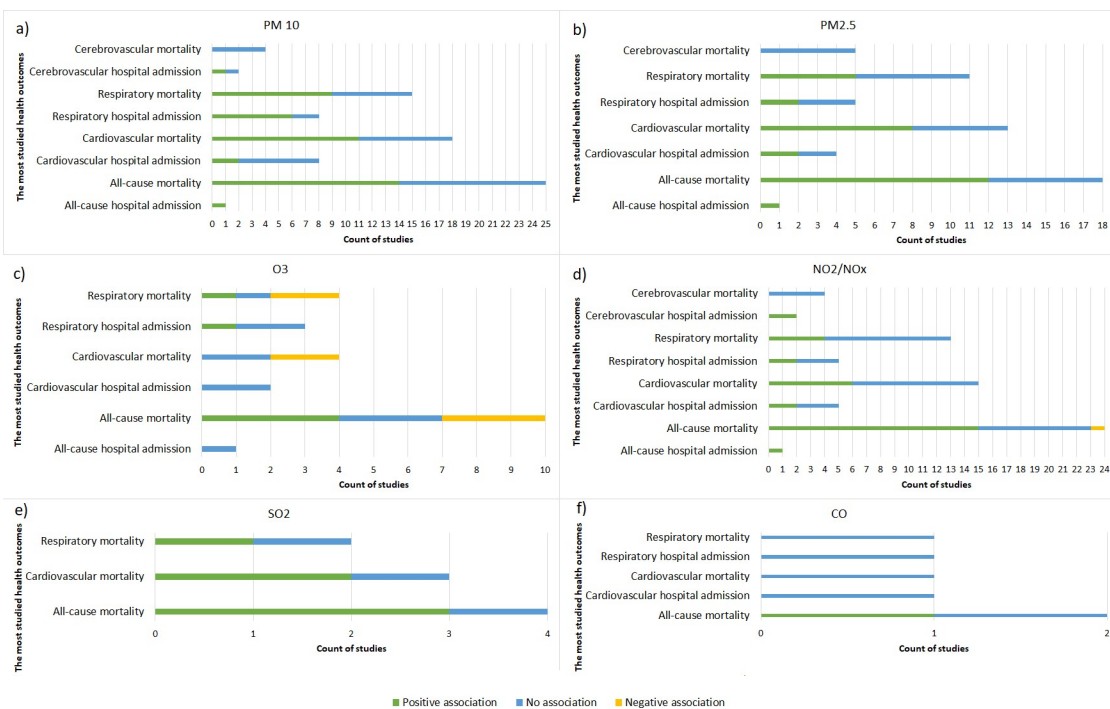

**Fig 3. The number of reviewed articles and the type of association between air pollutants and the most studied mortality and hospital admission outcomes.**

**The effect of ozone pollution on mortality and hospital admission.** Contrary to particulate matter pollution, the effect of ozone on mortality and hospital admission did not show a consistent effect. In some studies, ozone acted as a protective factor agianst mortality and hospital admission, while in other studies it showed increased risk or no association with mortality and hospital admissions (Fig 3C and S3 Table). This is related to the fact that ozone is a highly reactive pollutant and its formation is related to the presence of sunlight [25]. In a cohort study conducted by Carey et al. (2013) in England, ozone acted as a protective factor agianst all-cause mortality (HR = 0.96, 95%CI = 0.93 to 0.98), cardiovascular mortality (HR = 0.96, 95%CI = 0.94 to 0.98), respiratory mortality (HR = 0.93, 95%CI = 0.90 to 0.96), and lung cancer mortality (HR = 0.94, 95%CI = 0.90 to 0.98) [21]. However, ozone acted as a risk factor in some of the reviewed studies leading up to 2% increase in all-cause mortality per interquartile range increase of ozone concentration [34–37].

**The effect of nitrogen oxides pollution on mortality and hospital admission.** Similar to other air pollutants, this review showed that exposure to nitrogen dioxide and nitrogen oxides pollution can cause many types of diseases resulting in increased risk for all-cause mortality

**Table 2. The definitions of air temperature exposure classifications.**

| Classification | Definition |
| --- | --- |
| Cold temperature | Exposures to air temperature in the winter season below identified thresholds ranging from -7˚C to 6˚C |
| Hot temperature | Exposures to air temperature in the summer season above identified thresholds ranging from 20˚C to 37˚C |
| Air temperature increase | Exposures to increasing temperature across the whole year. Associations are interpreted per 1˚C increase in temperature. |

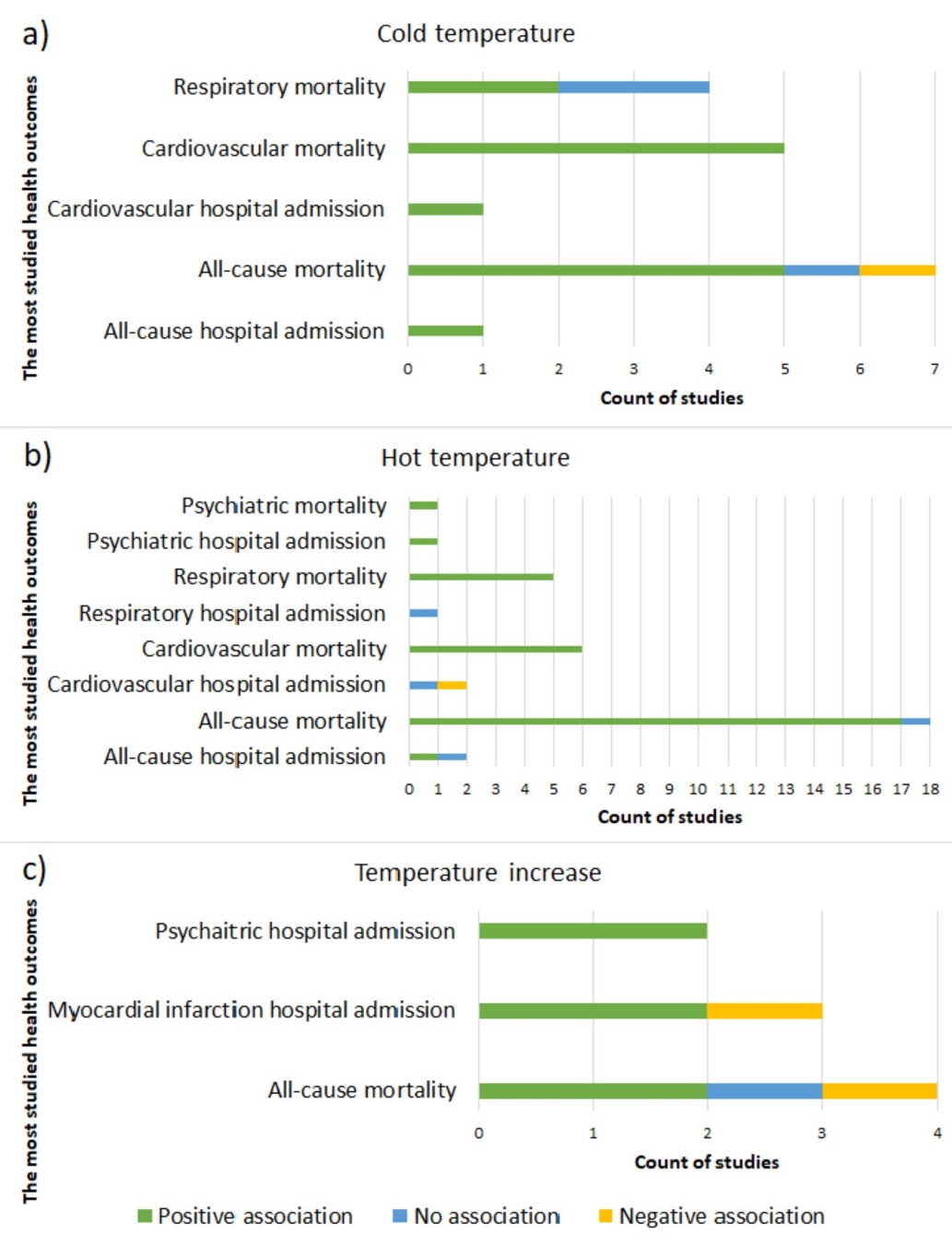

**Fig 4. The number of reviewed articles and the type of association between temperature and the most studied mortality and hospital admission outcomes.**

and hospital admission [25, 32, 38, 39] (Fig 3D and S3 Table). A study conducted in Belgium showed a 3.5% increase in cardiovascular hospital admission as well as 4.5% and 4.9% increase in ischemic stroke and haemorrhagic stroke hospital admissions respectively for each 10 μg/m3 increase in NO2 [40].

**The effect of sulphur dioxide pollution on mortality and hospital admission.** Sulphur dioxide air-pollution is mainly caused from industrial processes and power plants that involve

burning of fossil fuel. Exposure to SO2 pollution can cause mild health effects including eyes, nose, and throat irritations as well as severe health effects such as bronchial spasms and deaths due to respiratory insufficiency [41].

The effect of sulphur dioxide (SO2) on mortality and hospital admission was investigated in only 12 out of the 112 reviewed studies. Exposure to SO2 air-pollution was found to increase the risk for all-cause, cardiovascular, and respiratory mortality [21, 39, 42] (Fig 3E and S3 Table).

**The effect of carbon monoxide pollution on mortality and hospital admission.** Carbon monoxide results from incomplete combustion of fossil fuels. Carbon monoxide is dangerous for human beings since it possess the ability to bind to haemoglobin resulting in reduction of the red blood cells to carry oxygen to cells [41].

Only 10 out of the 112 reviewed studies investigated the association of carbon monoxide (CO) with mortality and hospital admission. The majority of these studies showed that carbon monoxide exposure can cause a number of cardiovascular and respiratory health problems (Fig 3F and S3 Table). Exposure to carbon monoxide pollution resulted in increased odds for pulmonary embolism hospital admission [43]. Additionally, Renzi et al. (2017) showed that all-cause mortality increases by 0.12% for every 1 mg/m3 increase in CO [39]. On the contrary, carbon monoxide acted as a protective factor against chest disease hospital admission among patients with sickle cell anaemia in one of the reviewed studies [44]. This association was explained by the fact that carbon monoxide can bind to haemoglobin which enhances the affinity of other binding sites for oxygen in addition to reducing vasoconstriction and inflammation; suggesting a beneficial effect rather than risk factor for patients with sickle cell disease [44].

## The effect of air temperature on mortality and hospital admission

Exposure to hot or cold temperature beyond region-specific thresholds exhibits a range of direct and indirect effects on human health. The direct effects include hyperthermia or heat stress during hot temperature exposures and hypothermia and ischemic stroke during cold temperature exposures [45]. Besides the direct effects, small fluctuations in temperature across time can result in indirect effects on the respiratory and cardiovascular systems of the body [45].

Most of the reviewed articles that studied the effect of weather exposure on mortality and hospital admission focused on air temperature exposure with lags ranging from 0 days up to 5 weeks for cold temperatures and from 0 days up to 25 days for hot temperatures. The reviewed studies examined the effect of cold temperature, hot temperature, and air temperature increase on a range of diseases, most commonly, cardiovascular, respiratory, and psychiatric disorders. Table 2 below shows the definitions of "cold temperature", "hot temperature", and "air temperature increase" classifications derived from the reviewed studies.

Cold temperature acted as a risk factor for several types of mortality and hospital admission outcomes (Fig 4A and S3 Table). Nevertheless, cold temperature was a protective factor only in one of the reviwed studies for all-cause mortality at lag 0 (RR = 0.99, 95%CI = 0.985 to 0.995); yet cold temperature acted as a risk factor for all-cause mortality in the same study at lag of 14 days with a relative risk of 1.003 emphasizing the delayed effect of cold temperature on mortality [46].

Similar to cold temperature, hot temperature also acted as a risk factor in most of the reviewed studies for a number of mortality and hospital admission outcomes (Fig 4B and S3 Table). On the other hand, hot temperatures were associated with reductions in hospital admission rates for ischemic heart disease (RR = 0.74, 95%CI = 0.55 to 0.99) in a study

conducted by Bijelovic et al. (2017) and for all-cause hospital admissions (RR = 0.961, 95% CI = 0.956 to 0.967) and cardiovascular hospital admissions (RR = 0.975, 95% CI = 0.957 to 0.993) in a study conducted by Monteiro et al. (2013) [47, 48].

Some studies examined the effect of increasing temperature across the whole year on mortality and hospital admission. More than half of these studies showed a significant positive association between the increasing temperature and the mortality and hospital admission outcomes (Fig 4C and S3 Table).

### The effect of other weather exposures on mortality and hospital admission

Similar to the temperature, weather exposures that include humidity, rainfall, sunshine, snow-cover, air pressure, daylight, wind speed and wind direction with lags ranging from 0 up to 7 days were found to affect a range of diseases, most commonly, cardiovascular, respiratory, and psychiatric disorders (S3 Table).

Weather variables that showed significant positive assciations with hospital admission included: a rainfall effect on psychiatric hospital admission [49], sunshine and daylight effects on hospital trauma [50] and psychaitric admissions [51], wind speed effects on chest disease hospital admission [44], and air pressure effects on mania and depression hospital admission [49].

It is worth mentioning that sunshine showed inconsistency in its effect on psychaitric hospital admission, acting as a risk factor in a Danish study [52] while acting as a protective factor in a study conducted in Ireland [49].

### The adjustments and effect modifications for the association of air-pollution and weather exposure with mortality and hospital admission

Most of the reviewed studies stratified and adjusted their analysis by age and gender [25, 40, 51, 53–61]. Socio-economic deprivation, education attainment, income level, marital status, and occupational class were considered as confounders or effect modifiers in some of the reviewed studies [25, 58, 62–66]. However, only one study considered ethnicity to act as an effect modifier in the association between all-cause mortality and "summer smog" days defined as having maximum temperature of 25˚C and PM10 pollutant oncentration of 50 µg/m3 [62]. And only two studies investigating the effect of air-pollution on all-cause and cardiovascular mortality in England adjusted for ethnicity in their multivariate regresison models [67, 68].

Some of the studies that examined the effect of air-pollution on mortality and hospital admission accounted for air temperature effect in their analysis [35, 40, 46]. Likewise, some of the reviewed articles that studied the association of weather exposure to mortality and hospital admission considered the effect of air-pollution in their analysis [34, 65, 69–72].

Other variables considered to affect the relationship of air-pollution and/or weather exposure with mortality and/or hospital admission included: weekend and holiday effect, population decrease during the summer, influenza epidemics, season, day of the week, and tobacco smoke [25, 40, 46, 54, 56, 73–79].

## Discussion

In this scoping review of 112 articles, we aimed to examine the effect of (1) air pollution, (2) temperature, and (3) other weather exposures on mortality and hospital admission outcomes.

The first part of the review showed that air-pollution acted consistently as a risk factor for all-cause, cardiovascular, respiratory, cerebrovascular and cancer mortality and hospital admission in the EU and UK which is in line with the findings of studies conducted in other

regions of the world [80–84]. For instance, elevated risks of cardiovascular and respiratory diseases mortality were reported in Istanbul-Turkey for every 10 μg/m3 increase in PM10, SO2 and NO2 pollutants [85]. An exception was ozone (O3) air-pollution which showed inconsistent association with mortality and hospital admission. Two explanations were offered in the literature for the negative association between health outcomes and ozone pollution. The first explanation is related to the fact that ozone is a highly seasonal pollutant since its formation is catalysed by sunlight rendering higher ozone concentrations in the summer as compared to winter season. Thus, ozone effect on health outcomes should be analysed by accounting for the season effect [21]. In the continental United States, a 49% higher risk in all-cause mortality was shown for every 10 ppb increase in ozone during the warm-season [86]. The second explanation is related to the high reactivity of ozone leading to the formation of other pollutants such as NO2 and particulate matter. Therefore, ozone is negatively correlated with other air pollutants and its effect on health outcomes should be analysed as a combined effect of O3 and NO2 (known as Ox effect) [25].

Additionally, our scoping review showed that the effect of particulate matter (PM10 and PM2.5) pollution on mortality and hospital admission is more studied in the literature as compared to the other air pollutants. This could be related to the more pronounced effects of particulate matter exposure on health which is corroborated by many studies across the world [81, 87]. Despite the fact that PM10 particles are deposited in the nasal cavities and upper airways, PM2.5 may penetrate deep into the lung tissues (reaching the alveoli and bloodstream) and irritate the respiratory airways causing various respiratory and cardiovascular problems [29, 30, 88].

Similar to air pollution, the second part of this review showed that hot and cold temperature exposures beyond region-specific thresholds are risk factors for a wide range of respiratory, cardiovascular (including: ischemic heart disease, myocardial infarction, pulmonary embolism, stroke, heart failure, and COPD), and psychiatric (including: mania and depression) illness in the EU and UK. These findings are corroborated by a wide body of literature from across the world [89–97]. In India, cold temperatures below 13.8˚C were associated with increased risk of 6.3% for all-cause mortality, 27.2% for stroke mortality, 9.7% for ischemic heart disease mortality, and 6.5% for respiratory diseases mortality [92]. In Istanbul-Turkey, 23 days of exposure to hot temperature above 22.8˚C was associated with a total of 419 excess deaths [90]. In Korea, hot temperature days of 25˚C compared to 15˚C were significantly associated with a 4.5% increase in cardiovascular hospitalizations [98].

It is worth to point out that the effect of cold temperature on health is more delayed (up to 5 weeks) in comparison to the more immediate effects of hot temperature (up to 25 days). Similar study in Northeast-Asia showed a delayed risk of cold temperature on mortality after 5 to 11 days, yet a more immediate effect of hot temperature on mortality after 1 to 3 days in each of Taiwan, Korea, and Japan countries [91].

Although exposure to hot or cold temperature can affect the health negatively, our scoping review showed that in few studies, the increase in temperature reduced the risk of hospital admissions for some types of cardiovascular diseases; mainly for pulmonary embolism, angina pectoris, chest, and ischemic heart diseases. This could be explained by the fact that hot temperature can cause immediate increase in cardiovascular mortality rates; whereby many cases might pass directly to the death state without passing through the hospital admission state resulting in lower hospital admission rates [47].

The third part of this scoping review presented the studies that examined the effects of other weather exposures such as relative humidity, barometric pressure, rainfall, and wind speed on mortality and hospital admission outcomes. These weather exposures were found to affect significantly only hospital admission. No significant effect was noted with respect to the

mortality outcome. The weather exposures acted as a risk factor for psychiatric disorders (including depression and Mania), chest disease, and trauma hospital admissions. This was corroborated by evidence from countries outside the EU as well [99–101]. Yet in some of the reviewed studies, weather exposures acted as a protective factor for some types of psychiatric and cardiovascular disorders. The significant negative association between ischemic heart disease hospital admission and humidity in one of the reviewed studies was explained by the fact that people in general and the elderly specifically reduce their activities during high humidity and temperature periods. This is mainly due to the lack of the body's ability to perspire, which in turn reduces their risk of cardiovascular complications [48]. As for the protective effect of some weather exposures on psychiatric hospital admissions, similar findings were presented in Iran; with a negative association between barometric pressure and schizophrenia hospital admissions and rainy days and bipolar hospital admissions [101].

In addition to the association of air pollution and weather exposure with mortality and hospital admission outcomes, our review aimed to present the individual, socio-economic, and environmental factors that play an important role in modifying the latter association. The effect modifiers identified in this scoping review included: pre-existing health conditions, age, gender, educational attainment, wealth or income or socio-economic deprivation, occupation, marital status, tobacco smoking, season, day of the week, holidays, and influenza epidemics.

Individuals with pre-existing chronic health conditions face increased susceptibility toward air-pollution and weather exposure related mortality and hospital admission [52, 54, 78, 102].

Older people are more vulnerable to the health effects associated with air-pollution, hot or cold temperatures, and other weather variables [54, 56, 103–105]. This is due to the physiological degeneration of the human body with increasing age. Aging affects the normal function of the body organs resulting in many chronic cardiovascular, urinary, and respiratory health conditions. This reduces the ability of older people to adapt to increased concentrations of air pollutants and changing weather conditions [103, 105, 106]. Moreover, old age people have lower immunity and antioxidant defence as compared to young people placing them at a higher risk [107]. Many older people also have reduced mobility and mental abilities which delay their access to healthcare leading to severe health complications and death [108].

As for gender, our review revealed inconsistency regarding its modification effect on the association between air-pollution and weather exposure and mortality and hospital admission health outcomes. Nevertheless, most of the reviewed studies have found that females have higher risks of mortality and/or hospital admission after exposure to air-pollution and/or weather fluctuations beyond region-specific thresholds including hot and cold temperatures [9, 47, 57, 58, 65, 78, 109–112]. Whereas some studies found higher risks of mortality and/or hospital admissions among males in relation to air-pollution and/or weather exposure [25, 40, 42, 59, 70, 113, 114]. One explanation for this might be due to the physiological differences between males and females. Females have smaller lung size, yet higher airways reactivity making them more susceptible to air-pollution health effects as compared to males [42, 115]. Likewise, higher pulse rates and smaller heart size relative to the human body in females as compared to males render females more vulnerable to the health effects of air-pollution and hot or cold temperature exposures [115]. Moreover, females exhibit more fluctuations in hormone levels due to pregnancy, menstrual cycle and menopause periods which may place them at a higher health risk upon exposure to air-pollution and weather variations [115]. The different lifestyle, socio-economic position, and occupation type between males and females may also lead to different levels and duration of air-pollution and weather exposure [62, 109, 116–118]. However, it is worth mentioning that the effect modification of gender in the association of air-pollution and weather exposure with mortality and hospital admission outcomes is believed to be confounded by age since in many of the reviewed studies, higher risks were

found among old aged females (age>65 years old) [47, 57, 61, 78, 112] and old aged males (age >70 years old) [114]. This confounding effect could be reduced either by assessing the combined effect modification of age and gender through an interaction term or by stratifying the analysis according to both the age groups and gender.

Wealth and socio-economic deprivation were also considered by some of the reviewed studies as an effect modifier in the relationship of air-pollution and weather with mortality and hospital admission. In general, the absence of wealth and presence of socio-economic deprivation increase the risk of exposure to air-pollution and weather variations resulting in elevated mortality and hospital admission rates in Europe [62, 66, 76] and in other parts of the world including New Zealand [22], United States of America [119], and Chile [120].

Educational attainment was also considered by some of the reviewed studies as an effect modifier, with higher risks detected among individuals with lower educational attainment [25, 63, 64, 121, 122]. Despite the consideration of age, gender, education, and wealth effect in the association of air-pollution and weather with mortality and hospital admissions in Europe, our scoping review revealed the lack of investigation into the role of other important socio-demographics such as ethnicity. Research has extensively shown that ethnic minorities live in more disadvantaged communities and have lower socio-economic status as well as poor housing conditions. This results in higher risk for chronic health problems associated with higher exposure on one hand and with lower access to quality healthcare on the other hand [22–24].

Finally, it is worth to note that most of the reviewed studies with a time-series or case-crossover design adjusted their analysis for the season effect [40, 46, 52, 54, 60, 71, 73, 74, 123, 124]. It is well established that air-pollution, temperature, and other weather variables vary with seasons [125–127]. Not to mention that the emission, formation, and dispersion of air pollutants is affected by seasonal weather variations which in turn affects the individual exposure levels [128]. Outdoor activities and daily habits (eg. Window ventilation of houses) might also vary depending on the season which reflect changes in the level and duration of individual exposure to air-pollution and weather changes [129].

Despite the value of this scoping literature review, it has some limitations. First, the employed search strategy was limited to original articles published in peer reviewed journals which might have led to the omission of unpublished work or articles that were published in non-indexed journals. Nevertheless, our search strategy involved navigation through two databases which enables a good catch of major published studies addressing the effect of air-pollution and weather exposure on mortality and hospital admission. Second, limiting our inclusion criteria only to English language articles might have resulted in missing some research written in other languages. However, as most of the literature worldwide is published in the English language, we believe that no major papers have been excluded. Third, this review was limited only to quantitative research which would have led to missing out other type of important research including opinion research pieces and letters to editor as well as qualitative research studies. Opinion research pieces and letters to editor provide a critical appraisal/discussion for the findings of original studies which warrant future research development. Qualitative studies provide an overview about the effect of air-pollution or weather variations on human health from the perspective of lay people rather than relying only on objective census/ statistics numbers as in quantitative research. Forth, due to resources limitations, title and abstract screening as well as data abstraction were done only by one researcher (MA). Nevertheless, a second researcher performed title and abstract screening for a random sample of 20% of the retrieved records. Given the high consensus between the two researchers, we are confident of the exact application of the inclusion and exclusion criteria. Our goal from this literature review was not to produce a numerical estimate but rather to give a narrative summary

on the effect of air-pollution/weather on mortality/hospital admission. Hence, missing some studies would not be a major concern for this scoping review.

### Literature gaps and implications for future research

This scoping review helped us to identify literature gaps that require further research.

First, this review revealed the extensive research carried out to determine the effect of air-pollution on human health. Yet, due to the high correlation between air pollutants and the issue of collinearity in multivariate models, most of the studies examined the effect of single pollutants on mortality and hospital admission outcomes. Nevertheless, the issue of correlation between air pollutants is highly contextual and it depends on the study settings including the season and the specific geographical area. Hence, future researchers should try to examine the effect of multi-pollutants on mortality and/or hospital admission in one model, where strong correlations between the air pollutants are absent.

Second, the majority of studies examined the direct effects of air-pollution and weather exposure on mortality and hospital admission without considering the role of certain effect modifiers. The examined effect modifiers considered mostly by the literature involve age, gender, education, socio-economic deprivation, and season. Therefore, there is a lack of evidence regarding the modifying effect of other individual factors such as previous disease conditions and ethnicity which affect the person's health vulnerability. Indeed, future research is needed to find out the reasons behind elevated individual's susceptibility to the detrimental effects of air-pollution and weather variations in certain groups of population.

Third, our review showed that most of the studies either investigated the effect of air-pollution or the effect of weather on mortality and hospital admission. The formation and dispersion of air pollutants depends highly on the existing atmospheric conditions such as temperature, humidity, and wind speed [130]. Therefore, future studies should consider examining the effect of both, weather conditions and air-pollution, on human health through interaction terms or adjustments in the analysis models.

Fourth, although extensive research has been performed to study the effect of particulate matter and nitrogen oxides pollution on human health, there was a lack of research with respect to other air pollutants including carbon monoxide, ozone, and sulphur dioxide. This might be due to the absence of rigorous and reliable measurements of these pollutants or due to the complexity of analysing the effect of these pollutants.

Fifth, literature is more focused on examining the effect of temperature on mortality and hospital admission, placing less emphasis on other weather exposures. Hence, future research should shift the focus toward other weather exposures such as wind speed, rainfall, humidity, snow cover, daylight, and air pressure.

Sixth, there was a lack of research examining the effect of air-pollution and weather on hospital admission. Mortality was the major outcome in most of the reviewed studies due to the ease of data access governed by less ethical considerations. Additionally, analysis is more straightforward given that it occurs only once in an individual's life. Thus, it is recommended for future research to consider the impact of air-pollution and weather variables on hospital admission on its own and in combination with mortality through multistate modelling.

Finally, the majority of studies in this field employ the time-series design which uses aggregated mortality and hospital admission data linked to environmental exposures at the local authorities or municipalities level. Research that uses aggregated data neglect the physiological and socio-economic differences among individuals. Additionally, assigning air-pollution and weather exposure based on wide geographies overlook the small geographical exposure differences biasing the drawn estimates. Therefore, there is a need for cohort research studies that

utilize individual level data linked to air-pollution and weather exposure at small geographical spatial resolution (eg. Postcodes).

## Conclusion

In summary, our scoping review showed that air-pollution and weather exposure beyond certain thresholds lead to various impacts on human health, most commonly cardiovascular and respiratory problems, resulting in increased rates of mortality and hospital admission. Yet, further research is needed given that the effect modification of important socio-demographics such as ethnicity and the interaction between air-pollution and weather is often missed in the literature. Understanding this should give enough evidence to the policy makers to plan and act accordingly aiming to reduce the effects of air pollution and weather variations on the public health. Additionally, research should focus on projecting future health behaviour and mortality patterns in relation to air pollution and weather variations, in order to guide effective environmental and health precautionary measures planning.

## Supporting information

**S1 Checklist. PRISMA checklist followed for this systematic scoping review data searching, screening, and abstraction.**
(DOC)

**S1 Table. The search codes used in PubMed and Web of Science databases for this scoping literature review.**
(DOCX)

**S2 Table. A detailed summary of the characteristics of the 112 studies included in this scoping review by the type of investigated health outcome.**
(DOCX)

**S3 Table. A detailed summary of the reported associations between air pollution and/or weather exposures and mortality and/or hospital admission outcomes in the 112 reviewed studies in terms of coefficients and 95% confidence intervals.**
(DOCX)

## Acknowledgments

The authors would like to thank the University of St Andrews Library services for helping in developing the search codes used in this scoping literature review.

## Author Contributions

**Conceptualization:** Mary Abed Al Ahad, Frank Sullivan, Urška Demšar, Hill Kulu.

**Data curation:** Mary Abed Al Ahad.

**Formal analysis:** Mary Abed Al Ahad.

**Funding acquisition:** Frank Sullivan, Urška Demšar, Hill Kulu.

**Investigation:** Mary Abed Al Ahad, Hill Kulu.

**Methodology:** Mary Abed Al Ahad, Maya Melhem.

**Project administration:** Mary Abed Al Ahad.

**Resources:** Mary Abed Al Ahad.

**Supervision:** Frank Sullivan, Urška Demšar, Hill Kulu.

**Validation:** Maya Melhem.

**Visualization:** Mary Abed Al Ahad.

**Writing – original draft:** Mary Abed Al Ahad.

**Writing – review & editing:** Mary Abed Al Ahad, Frank Sullivan, Urška Demšar, Hill Kulu.

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
