## [Decision Letter · Decision Letter 0]

25 Aug 2020

PONE-D-20-19327

The Effect of Air-pollution and Extreme-weather on Mortality and Hospital Admission and implications for further research: A Systematic Scoping Review

PLOS ONE

Dear Dr. Abed Al Ahad,

Thank you for submitting your manuscript to PLOS ONE. After careful consideration, we feel that it has merit but does not fully meet PLOS ONE’s publication criteria as it currently stands. Therefore, we invite you to submit a revised version of the manuscript that addresses the points raised during the review process.

We look forward to receiving your revised manuscript.

Kind regards,

Chon-Lin Lee, Ph.D.

Academic Editor

PLOS ONE

Journal Requirements:

2. Your ethics statement must appear in the Methods section of your manuscript. If your ethics statement is written in any section besides the Methods, please move it to the Methods section and delete it from any other section. Please also ensure that your ethics statement is included in your manuscript, as the ethics section of your online submission will not be published alongside your manuscript.

Reviewers' comments:

Reviewer's Responses to Questions

**Comments to the Author**

1. Is the manuscript technically sound, and do the data support the conclusions?

Reviewer #1: Partly

Reviewer #2: Yes

2. Has the statistical analysis been performed appropriately and rigorously? 

Reviewer #1: Yes

Reviewer #2: N/A

3. Have the authors made all data underlying the findings in their manuscript fully available?

Reviewer #1: Yes

Reviewer #2: Yes

4. Is the manuscript presented in an intelligible fashion and written in standard English?

Reviewer #1: Yes

Reviewer #2: Yes

5. Review Comments to the Author

Reviewer #1: The manuscript entitled “The Effect of Air-pollution and Extreme-weather on Mortality and Hospital Admission and implications for further research: A Systematic Scoping Review” used the scoping review approach to summarize the literature on the association of air pollution and extreme weather with mortality and hospital admissions. The authors used the reviews to conclude that air pollution and extreme weather affect human health negatively. They also highlighted the literature gaps that require further research. This manuscript addresses an interesting environmental health issue and contains some useful information; however, I have several major concerns that need to be adequately addressed before the manuscript can be considered for publication.

1. The author should clearly define “Extreme-weather.” It is commonly understood that extreme weather is when weather is significantly different from the usual weather pattern. However, most publications discuss the effect of ambient temperature, relative humidity, or other meteorological factors on health effects. The authors need to clarify the terminology. Also, modification of the title is also suggested.

2. Lines 42, the author mention “Climate Change” and briefly described it. Nevertheless, the effects of climate change are not discussed in the manuscript. The authors could consider to delete it in the background section.

3. Line 53-55, the author state that “there is a lack of information on the role of some effect modifiers such as ethnicity and the interaction between air-pollution and extreme-weather.” I suggest that the author provide some evidence about the importance of effect modifiers (ethnicity) and the interaction between air-pollution and extreme-weather (meteorological factors) in the context of their health effects.

4. Particulate matter (PM2.5 or PM10) is heterogeneous mixtures of solid and liquid particles emitted from a variety of sources. Recently, there are Along with size, concentration, and chemical components of particulate matter are important in mediating the effects of PM on human health. Although the evidence of PM composition with those adverse health effects is limited, I believe that, in this review, this issue is worth mentioning in the background.

5. The reasons for excluding the pediatric population is not justified. For assessing health impacts, to evaluate the whole population (from pediatrics to geriatrics) is very important. Especially, the authors attempt to provide a comprehensive review of the topic. I highly suggest the author included the pediatric population.

6. Line 132, one of the exclusion criteria is “studies investigating in-hospital death.” Is in-hospital death part of overall mortality? Why exclude those publications?

7. Line 212 and 377, the author state that “particulate matter especially the small-size ones (PM10 and PM2.5) penetrate deeply the respiratory system” is not entirely correct. Most PM10 particles are deposited in the nasal cavities and upper airways. However, PM2.5 particles may penetrate the lung alveoli and enter into the bloodstream. (Möller W, Felten K, Sommerer K, Scheuch G, Meyer G, Meyer P, et al. Deposition, retention, and translocation of ultrafine particles from the central airways and lung periphery. Am J Respir Crit Care Med. 2008;177(4):426–32.)

8. Line 236-238 (This is related to the fact that ozone is a highly reactive pollutant and its formation is related to the presence of sunlight), the citation is needed for the statement.

9. In section 3.2. (The effect of air temperature on mortality and hospital admission), the cold, warm, and hot temperatures are needed to be clearly defined. Is the temperature cutoff identical among the references?

10. Line 528, the author discussed the correlation between air pollutants. In multipollutant models, because variables are commonly highly correlated, the collinearity becomes the major problem for multivariate analysis. I suggested the author describe the issue briefly.

11. In the conclusion section (line 571 and 573), the authors mention “climatic change.” However, this review does not touch upon the topic (climatic change). Overall, the sections “Discussion” is not well-organized and well-presented. It needs to be significantly revised.

Reviewer #2: This is a review paper that summarized 106 published works on air pollution and weather on mortality and hospital admission. The work followed PRISMA guideline to search and screen from literature.

Major points:

1. the goal of this manuscript is stated at line 510-512, not a numerical estimate but a narrative summary, these words should be addressed at the abstract or introduction as well.

2. a little bit confused at 4.2 literature gap part. The first suggestion stated that the exact role of individual pollutants is still unclear; but the third said that most studies examined the effect of single pollutants. The two statements contradict to each other. It might be better to merge the two gaps into one, and emphasize the interaction of the variables is the missing link.

Minor points:

1. Line 116, typo: heat.

2. Line505, missing a space between first two words.

3. Figures 3 and 4, please indicate what the y-axis is.

6. PLOS authors have the option to publish the peer review history of their article (what does this mean?). If published, this will include your full peer review and any attached files.

Reviewer #1: No

Reviewer #2: No

---

## [Author Response · Author response to Decision Letter 0]

15 Sep 2020

Professor Chon-Lin Lee 

Academic Editor

PLOS ONE

Manuscript title: The Effect of Air-pollution and Extreme-weather on Mortality and Hospital Admission and implications for further research: A Systematic Scoping Review

PONE-D-20-19327

**Response to the Editor’s and Reviewers’ Comments 

We thank the editor and the reviewers for their encouraging feedback and this opportunity to improve our manuscript. Please find below a point by point answer to the reviewers’ comments and the necessary changes made to the manuscript in track-changes. We also submitted a clean version of the manuscript in addition to the track-changes version. 

Thank you for considering our work and we hope that the revised version is suitable for publication and we are looking forward to hearing from you again.

Sincerely,

 Mary Abed Al Ahad, on behalf of the authors

Journal Requirements

**Authors’ response: We have abided by the journal requirements during the submission of this version of the manuscript in terms of following the Vancouver style for citation and the journal’s style for headings/subheadings and tables and figures.

2. Your ethics statement must appear in the Methods section of your manuscript. If your ethics statement is written in any section besides the Methods, please move it to the Methods section and delete it from any other section. Please also ensure that your ethics statement is included in your manuscript, as the ethics section of your online submission will not be published alongside your manuscript.

**Authors’ response: Ethics statement is not applicable for this manuscript as it is a scoping review of literature that is publicly available on “Pubmed” and “Web of Science” search engines. 

**Authors’ response: We included captions for the supporting information files at the end of the revised manuscript and updated the in-text citations to match accordingly. 

Reviewers’ comments

Reviewer #1: 

The manuscript entitled “The Effect of Air-pollution and Extreme-weather on Mortality and Hospital Admission and implications for further research: A Systematic Scoping Review” used the scoping review approach to summarize the literature on the association of air pollution and extreme weather with mortality and hospital admissions. The authors used the reviews to conclude that air pollution and extreme weather affect human health negatively. They also highlighted the literature gaps that require further research. This manuscript addresses an interesting environmental health issue and contains some useful information; however, I have several major concerns that need to be adequately addressed before the manuscript can be considered for publication.

1. The author should clearly define “Extreme-weather.” It is commonly understood that extreme weather is when weather is significantly different from the usual weather pattern. However, most publications discuss the effect of ambient temperature, relative humidity, or other meteorological factors on health effects. The authors need to clarify the terminology. Also, modification of the title is also suggested.

**Authors’ response: We have replaced the term “extreme weather” by “weather exposure” in the manuscript as our objective was to provide an overview on articles discussing the effect of ambient temperature, relative humidity and other weather exposures which could not merely be extreme weather exposures. We have also provided examples to explain what we mean by “weather exposure” in the introduction section on line 43 “Weather exposure in terms of changing temperature, relative humidity, rainfall and other weather patterns can cause a wide range of acute illness …..”. 

We modified the title to be in line with the new terminology of “weather exposure”. The new title is: “The Effect of Air-pollution and weather exposure on Mortality and Hospital Admission and implications for further research: A Systematic Scoping Review”. 

2. Lines 42, the author mention “Climate Change” and briefly described it. Nevertheless, the effects of climate change are not discussed in the manuscript. The authors could consider deleting it in the background section.

**Authors’ response: We have deleted the “climate change” phrase from the Introduction section. 

3. Line 53-55, the author state that “there is a lack of information on the role of some effect modifiers such as ethnicity and the interaction between air-pollution and extreme-weather.” I suggest that the author provide some evidence about the importance of effect modifiers (ethnicity) and the interaction between air-pollution and extreme-weather (meteorological factors) in the context of their health effects.

**Authors’ response: Evidence about the importance of effect modifiers (ethnicity) and the interaction between air-pollution and weather exposure in the context of their health effects was added to the introduction section of the manuscript on page 4, line 55 to 65 as follows: “Though, there is a lack of information on wider aspects including the role of some effect modifiers such as ethnicity and the interaction between air-pollution and weather factors. Literature has shown that ethnic minorities often live in more disadvantaged, highly populated urban communities with poor housing conditions and higher levels of air pollution exposure (22-24). This results in poorer health and higher risk for chronic health problems with time. Similar to ethnicity, the interaction between air-pollution and weather variables in relation to health outcomes is often missed in the literature despite its importance in minimizing biased estimations. Air pollutants are highly reactive, and their formation is either catalyzed or slowed down based on the existing weather conditions. For example, the presence of sunlight catalyzes the formation of ozone pollutant resulting in higher ozone concentrations during the summer (25)”. 

4. Particulate matter (PM2.5 or PM10) is heterogeneous mixtures of solid and liquid particles emitted from a variety of sources. Recently, there are Along with size, concentration, and chemical components of particulate matter are important in mediating the effects of PM on human health. Although the evidence of PM composition with those adverse health effects is limited, I believe that, in this review, this issue is worth mentioning in the background.

**Authors’ response: We included in the manuscript on page 11, line 216 to 226 a brief description about the effect of particulate matter on human health which is related to their size, composition, and concentration as follows: “Particulate matter is a heterogeneous mixtures of liquid droplets and solid particles suspended in the air that can result either from natural resources (windblown Saharan and non-Saharan dust, volcano ashes, forest fires, pollen, etc…) or from man-made activities including industrial processes, transportation vehicle smoke, burning of fossil fuels, extensive energy usage, combustion processes, and grinding and mining industries (28). Due to its size, mass composition, and chemical components, particulate matter with larger diameter ……”. 

We mentioned this on page 11, line 216-226 in the section where we are talking about the effect of particulate matter on human health as it fits more the flow of the manuscript than including it in the introduction section. 

5. The reasons for excluding the pediatric population is not justified. For assessing health impacts, to evaluate the whole population (from pediatrics to geriatrics) is very important. Especially, the authors attempt to provide a comprehensive review of the topic. I highly suggest the author included the pediatric population.

**Authors’ response: We have added the studies on pediatrics population (4 studies in total) to our scoping review which brings the total of reviewed studies from 106 to 110 studies. Below are the references for these 4 studies:

• Ghirardi, L., Bisoffi, G., Mirandola, R., Ricci, G., & Baccini, M. (2015). The Impact of Heat on an Emergency Department in Italy: Attributable Visits among Children, Adults, and the Elderly during the Warm Season. PLoS One, 10(10). doi:10.1371/journal.pone.0141054

• Janke, K. (2014). Air pollution, avoidance behaviour and children's respiratory health: Evidence from England. Journal of Health Economics, 38, 23-42. doi:10.1016/j.jhealeco.2014.07.002

• Litchfield, I. J., Ayres, J. G., Jaakkola, J. J. K., & Mohammed, N. I. (2018). Is ambient air pollution associated with onset of sudden infant death syndrome: a case-crossover study in the UK. BMJ Open, 8(4). doi:10.1136/bmjopen-2017-018341

• Piel, F. B., Tewari, S., Brousse, V., Analitis, A., Font, A., Menzel, S., . . . Rees, D. C. (2017). Associations between environmental factors and hospital admissions for sickle cell disease. Haematologica, 102(4), 666-675. doi:10.3324/haematol.2016.154245

6. Line 132, one of the exclusion criteria is “studies investigating in-hospital death.” Is in-hospital death part of overall mortality? Why exclude those publications?

**Authors’ response: We have added the studies addressing in-hospital death as part of overall mortality (2 studies in total) to our scoping review which brings the total of reviewed studies from 110 to 112 studies. Below are the references for these 2 studies:

• Callaly, E., Mikulich, O., & Silke, B. (2013). Increased winter mortality: the effect of season, temperature and deprivation in the acutely ill medical patient. Eur J Intern Med, 24(6), 546-551. doi:10.1016/j.ejim.2013.02.004

• Lyons, J., Chotirmall, S. H., O'Riordan, D., & Silke, B. (2014). Air quality impacts mortality in acute medical admissions. Qjm, 107(5), 347-353. doi:10.1093/qjmed/hct253

7. Line 212 and 377, the author state that “particulate matter especially the small-size ones (PM10 and PM2.5) penetrate deeply the respiratory system” is not entirely correct. Most PM10 particles are deposited in the nasal cavities and upper airways. However, PM2.5 particles may penetrate the lung alveoli and enter into the bloodstream. (Möller W, Felten K, Sommerer K, Scheuch G, Meyer G, Meyer P, et al. Deposition, retention, and translocation of ultrafine particles from the central airways and lung periphery. Am J Respir Crit Care Med. 2008;177(4):426–32.)

**Authors’ response: We have modified the text on line 220-226 and line 386-393 in the new version of the manuscript to reflect the fact that PM10 particles are deposited in the nasal cavities and that PM2.5 can penetrate deep the lungs reaching the alveoli and blood stream and added the above reference for the statement. 

8. Line 236-238 (This is related to the fact that ozone is a highly reactive pollutant and its formation is related to the presence of sunlight), the citation is needed for the statement.

**Authors’ response: Sorry for missing out the citation for that statement. We have now added citation “citation number 25” to this statement on line 246 of the new version of the manuscript. 

9. In section 3.2. (The effect of air temperature on mortality and hospital admission), the cold, warm, and hot temperatures are needed to be clearly defined. Is the temperature cutoff identical among the references?

**Authors’ response: We have added a table “Table 2” which includes definitions for hot and cold temperature exposures with a range of cutoff (threshold) points. We took the classification of “hot” or “cold” temperature from the studies themselves and each study had its own identified cutoff point for hot and/or cold temperature. 

Table 2. The definitions of air temperature exposure classifications

Classification Definition 

Cold temperature Exposures to air temperature in the winter season below identified thresholds ranging from -7 ºC to 6 ºC

Hot temperature Exposures to air temperature in the summer season above identified thresholds ranging from 20 ºC to 37 ºC

Air temperature increase Exposures to increasing temperature across the whole year. Associations are interpreted per 1 ºC increase in temperature. 

10. Line 528, the author discussed the correlation between air pollutants. In multipollutant models, because variables are commonly highly correlated, the collinearity becomes the major problem for multivariate analysis. I suggested the author describe the issue briefly.

**Authors’ response: We have now described the issue briefly and amended the paragraph on line 525 to 531 in the revised version of the manuscript as follows: “Yet, due to the high correlation between air pollutants and the issue of collinearity in multivariate models, most of the studies examined the effect of single pollutants on mortality and hospital admission outcomes. Nevertheless, the issue of correlation between air pollutants is highly contextual and it depends on the study settings including the season and the specific geographical area. Hence, future researchers should try to examine the effect of multi-pollutants on mortality and/or hospital admission in one model, where strong correlations between the air pollutants are absent”. 

11. In the conclusion section (line 571 and 573), the authors mention “climatic change.” However, this review does not touch upon the topic (climatic change). Overall, the sections “Discussion” is not well-organized and well-presented. It needs to be significantly revised.

**Authors’ response: We have revised the conclusion section and replaced “climatic change” with “air pollution and weather variations” on line 577 in the revised manuscript. 

Additionally, we have revised extensively the “Discussion” section to be more organized and well-presented which could be viewed in the track-changes version of the revised manuscript. 

Reviewer #2: 

This is a review paper that summarized 106 published works on air pollution and weather on mortality and hospital admission. The work followed PRISMA guideline to search and screen from literature.

Major points:

1. the goal of this manuscript is stated at line 510-512, not a numerical estimate but a narrative summary, these words should be addressed at the abstract or introduction as well.

**Authors’ response: We have added that the scoping review aimed for a narrative summary of the literature in both, the introduction (on line 71) and the abstract (on line 5). 

2. a little bit confused at 4.2 literature gap part. The first suggestion stated that the exact role of individual pollutants is still unclear; but the third said that most studies examined the effect of single pollutants. The two statements contradict to each other. It might be better to merge the two gaps into one, and emphasize the interaction of the variables is the missing link.

**Authors’ response: Thank you for the comment. We have now merged the first and third literature gaps into one, focusing on the correlation and interaction between the air pollution variables on line 524-531 of the revised manuscript as follows: “First, this review revealed the extensive research carried out to determine the effect of air-pollution on human health. Yet, due to the high correlation between air pollutants and the issue of collinearity in multivariate models, most of the studies examined the effect of single pollutants on mortality and hospital admission outcomes. Nevertheless, the issue of correlation between air pollutants is highly contextual and it depends on the study settings including the season and the specific geographical area. Hence, future researchers should try to examine the effect of multi-pollutants on mortality and/or hospital admission in one model, where strong correlations between the air pollutants are absent”. 

Minor points:

1. Line 116, typo: heat.

**Authors’ response: The typo mistake “heath” on line 116 was corrected to “heat”. 

2. Line505, missing a space between first two words.

**Authors’ response: The space between first two words on line 505 was corrected.

3. Figures 3 and 4, please indicate what the y-axis is.

**Authors’ response: We have added the Y-axis and the X-axis descriptions to Figures 3 and 4

---

## [Decision Letter · Decision Letter 1]

15 Oct 2020

The Effect of Air-pollution and weather exposure on Mortality and Hospital Admission and implications for further research: A Systematic Scoping Review

PONE-D-20-19327R1

Dear Dr. Abed Al Ahad,

We’re pleased to inform you that your manuscript has been judged scientifically suitable for publication and will be formally accepted for publication once it meets all outstanding technical requirements.

Kind regards,

Chon-Lin Lee, Ph.D.

Academic Editor

PLOS ONE

Additional Editor Comments (optional):

Reviewers' comments:

Reviewer's Responses to Questions

**Comments to the Author**

1. If the authors have adequately addressed your comments raised in a previous round of review and you feel that this manuscript is now acceptable for publication, you may indicate that here to bypass the “Comments to the Author” section, enter your conflict of interest statement in the “Confidential to Editor” section, and submit your "Accept" recommendation.

Reviewer #1: All comments have been addressed

Reviewer #2: All comments have been addressed

2. Is the manuscript technically sound, and do the data support the conclusions?

Reviewer #1: Yes

Reviewer #2: (No Response)

3. Has the statistical analysis been performed appropriately and rigorously? 

Reviewer #1: Yes

Reviewer #2: (No Response)

4. Have the authors made all data underlying the findings in their manuscript fully available?

Reviewer #1: Yes

Reviewer #2: (No Response)

5. Is the manuscript presented in an intelligible fashion and written in standard English?

Reviewer #1: Yes

Reviewer #2: (No Response)

6. Review Comments to the Author

Reviewer #1: Thank you for having me to review the article entitled “The Effect of Air-pollution and weather exposure on Mortality and Hospital Admission and implications for further research: A Systematic Scoping Review.” This manuscript addresses an interesting environmental health issue. The authors put great effort into revising the manuscript. New, the article is well-written and contained important information to the knowledge domain about the health effects of air pollution. I think it is worthy of being published.

Reviewer #2: (No Response)

7. PLOS authors have the option to publish the peer review history of their article (what does this mean?). If published, this will include your full peer review and any attached files.

Reviewer #1: No

Reviewer #2: No

---

## [Editor Report · Acceptance letter]

19 Oct 2020

PONE-D-20-19327R1 

The Effect of Air-pollution and weather exposure on Mortality and Hospital Admission and implications for further research: A Systematic Scoping Review  

Dear Dr. Abed Al Ahad:

I'm pleased to inform you that your manuscript has been deemed suitable for publication in PLOS ONE. Congratulations! Your manuscript is now with our production department. 

Kind regards, 

on behalf of

Dr. Chon-Lin Lee 

Academic Editor

PLOS ONE